# Embodiment modifies attention allotment for the benefit of dual task performance

Yukiko Iwasaki [1✉], Benjamin Navarro[2], Hiroyasu Iwata[1] & Gowrishankar Ganesh [2✉]

Many everyday tasks, like walking down a street, require us to dual task to also avoid collisions of our swinging arms with other pedestrians. The collision avoidance is possible with ease because humans attend to all our (embodied) limbs. But how does the level of embodiment affect attention distribution, and consequently task performance in dual tasks? Here we examined this question with a dual task that required participants to perform a cued button-press (main task) with their right hand, while reacting to possible collisions by a moving object with a left 'robot' hand (secondary task). We observed that participants consistently improve main task performance when they perceived the robot hand to be embodied, compared to when they don't. The secondary task performance could be maintained in both cases. Our results suggest that embodiment of a limb modifies attention allotment for the benefit of dual motor task performance using limbs.

[1] Graduate School of Creative Science and Engineering, Waseda University, Shinjuku, Tokyo 1628480, Japan. [2] Laboratoire d'Informatique, de Robotique et de Microelectronique de Montpellier (LIRMM), University Montpellier, CNRS, Montpellier 34095, France. ✉email: kamiwaza@ruri.waseda.jp; ganesh.gowrishankar@lirmm.fr

Humans perceive a sense of "bodily self-consciousness"[1,2] or "embodiment"[3,4] toward their limbs. Embodiment of a limb is believed to include a sense of "ownership", a sense of an ability to control, or "agency", and a sense of "location" of the limb[4]. Ownership of a limb is known to improve its visual awareness[5], But it remains unclear how embodiment of one limb modulates the attention distribution across limbs.

Attention distribution across limbs is essential in many daily life movement tasks. This is because limb movements often require us to dual-task[6] to also avoid collisions by the rest of our body with environmental objects and other humans. For example, imagine you are walking down a supermarket aisle. Even though the focus of your attention may be on picking the grocery items with one hand, you need to still, avoid colliding with the other shoppers, with your other swinging arm. Here, we investigate the effect of embodiment on such dual tasks. Specifically, we are interested to understand how the level of embodiment of the secondary task (collision) performing arm, modulates the attention distribution between the arms, and consequently, the main task (grocery picking in the above example) performance by an individual.

To address this issue, we developed a dual task in virtual reality (VR) motivated by the above shopping example. Our task required participants to perform a visually cued button-press task with their right hand (their main task which required high attention) while reacting to possible collisions by a moving object that sometimes approached their left *robot* arm (the secondary, low attention, collision avoidance task). Recent studies have shown that the human self is plastic and that multi-sensory stimulation can induce a sense of embodiment in humans, toward a rubber hand[7–11] as well as functionally similar[12] robot limbs[13,14]. Here, we used multi-sensory stimulations to modulate the sense of embodiment perceived by the participant towards their left robot arm and created two conditions in which the perception of embodiment of the robot limb was different. We then investigated how the embodiment (measured using a behavioral measure and subjective reports in a questionnaire) affects the performance of the main task performed by the right hand. We chose to use a robotic left arm to avoid any pre-existing attentional biases associated with the shape of the human arm.

We hypothesized one of two possible scenarios. If increased embodiment of the robot arm attracts more attention toward the robot arm, then this will be evident as a decrease in performance in the right-hand main task. Conversely, an increase in right-hand performance is expected if embodiment of the robot arm either enables increased attention to the right (main task) arm, or improves the attention distribution between the two arms.

## Results

Our experiment required participants to wear a VR headset and hold a haptic feedback device (Haption Virtuose 3D) in their left hand (Fig. 1a, lower). They were shown a virtual right hand in the same location as their right hand. The virtual right fingers were shown placed on a keyboard, corresponding to their own fingers so as to synchronize the visuohaptic stimulation of the keyboard touch. The participants were presented with a robot arm in place of their real left arm (Fig. 1a, upper). The robot hand was purposely presented displaced, by 10 cm horizontally towards the body midline, from the real hand. This displacement was utilized to quantify the *proprioceptive drift*[4,7,15,16] toward the left arm. Any movement of the left arm by the human was displayed as the movement of the robot inside the VR environment.

All participants performed in two conditions (the order was balanced across participants). In the robot embodiment (EMB) condition, after the calibration and setup in the initialization

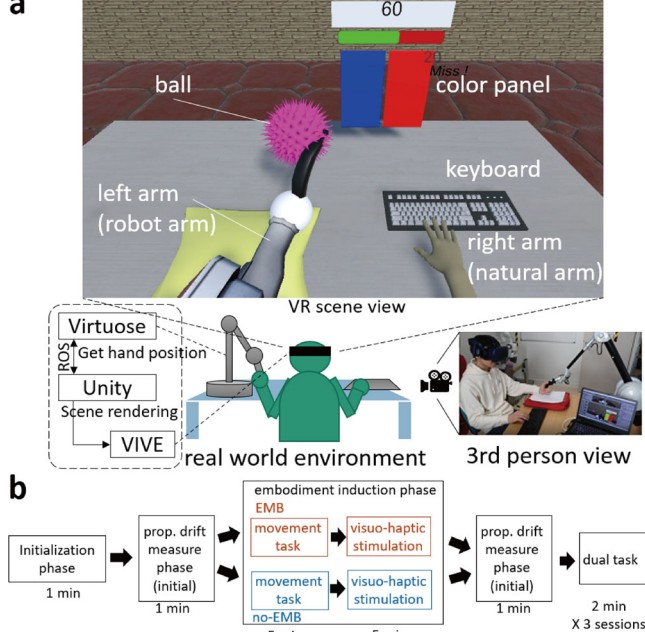

**Fig. 1 Setup and paradigm of the experiment. a** The participants performed the experiment in a virtual environment. They wore a head-mounted display and sat on a table with a keyboard under their right hand. They held the Virtuose haptic interface in their left hand. In the virtual environment, the participants observed a table and a right hand with the keyboard (upper panel). They saw a robot arm instead of their left arm. The robot left hand held a black banana-shaped object that was the shape of the handle of the haptic interface the participants held in their real left hand. The participants were also shown a pink ball which moved near the robot's hand/arm. **b** The participants worked in two conditions, EMB and no-EMB. Each condition consisted of five phases, and lasted 20 min in total.

phase, we induced the sense of embodiment (see Fig. 1b) toward the robot arm using standard visuohaptic stimulation techniques (see "Methods" for details). In the no-embodiment (no-EMB) condition, the same stimulations were presented asynchronously to prevent embodiment of the robot arm. The synchronized visuohaptic feedback of the keyboard button touch (by the real and virtual right finger) was present throughout both conditions. No other stimulation was presented to the real or virtual right arm. We utilized a proprioceptive localization task before and after the stimulation phase to evaluate proprioceptive drift as a possible behavioral measure of the induced embodiment on the left arm. This was then followed by the experimental dual task in each condition, which required the participants to perform a main task with their right hand and a secondary collision avoidance task with their left robot hand.

The participants were presented with a screen in their right visual field inside the VR, and in front of their right hand that rested on a keyboard. The main task required the participants to watch two rectangular panels on the screen, that changed their colors randomly every 500 ms. The participants were instructed to "press the space key as soon as the colors of the two rectangles became the same", which happened roughly every 2.5 s. We analyzed the *reaction time* of the participants, defined as the time between when the color of the rectangles became the same, and the button pressed by the participant.

The participants were presented with a pseudo-randomly flying ball (speed range: 0.25–0.75 m/s) in the left visual field in VR, that sometimes approached the left robot arm/arm of the participants. As their secondary task, the participants were required to press a *collision avoidance button* (ca-button) with their left thumb on

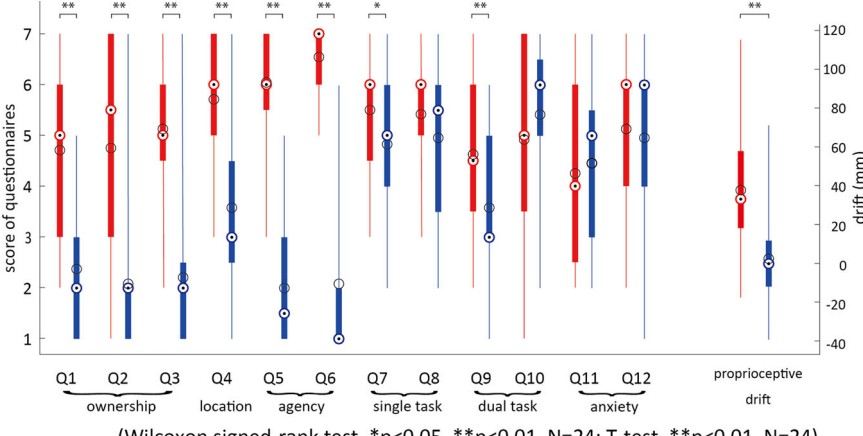

**Fig. 2 Embodiment modulation across conditions.** The participants scored twelve questions on a Likert scale after the EMB (red plot) and no-EMB (blue plot) conditions. The black circles show the mean and the colored circle with the dot shows the median scores across participants. The box edges show the 25th and 75th percentile of the data and the whiskers show the data range. The questions verified their subjective perception of ownership, location, the agency of the robot arm, their single- and dual-task performance, and the anxiety during their performance. We considered the average score from their first six questions as the measure of embodiment (the embodiment score) perceived towards the robot hand. We also measured the proprioceptive drift in each condition.

the handle of the haptic device held in their left hand when they perceived danger of collision. The ca-button press resulted in the ball being deflected away from their hand. The participants were asked not to move their left arm during the task. Crucially, the ball approached the hand every 5–8 s. Therefore, apriori, the main task required much higher attention compared to the secondary collision avoidance task.

The participants were provided with reward points for their dual-task performance. They were informed that the points correlated with the reaction speed of their presses, and that they will be penalized for erroneous presses (see "Methods" for details). Collisions with the left hand/arm resulted in heavy penalization of points. The main task performance (between the EMB and no-EMB conditions) was compared by the reaction time, while the secondary left-hand task performance was quantified by the number of left-hand collisions, and the distance between the hand and the ball, when the ca-button was pressed (see "Methods" for more details).

Note that the tasks on both arms involved only finger presses and did not require any hand or arm movement by the participants in either condition. We evaluated the cognitive sense of ownership, agency, location, and task performance using a questionnaire at the end of each condition.

Figure 2 shows the answers to the questionnaire, and the proprioceptive drift observed in the two conditions. The source data underlying Fig. 2 is provided as Supplementary data 1. We observed that the ownership (average score in Q1, Q2, Q3), left hand location (Q4) as well the sense of agency (average of Q5 and Q6) toward the robot arm were consistently higher $(Z(23) = 3.947, P < 0.0001; Z(23) = 3.584, P < 0.0001,$ and $Z(23) = 4.259, P < 0.0001$, respectively) in the EMB condition, in comparison to the no-EMB condition. Overall the robot arm embodiment (average of Q1 to Q6) was higher in the EMB condition, in comparison to the no-EMB condition $(Z(23) = 4.201, P < 0.0001)$. The participants also perceived higher left-hand task performance $(Q7, Z(23) = 2.275, P = 0.023)$, and dual-task performance $(Q9, Z(23) = 2.865, P = 0.005)$ in the EMB condition compared to no-EMB condition. Finally, the proprioceptive drift was observed to be higher in the EMB condition, compared to the no-EMB condition

$(T(23) = 4.7693, P < 0.001, d = 1.1897$, T-test), even though none of the participants reported noticing this shift when asked after the end of the entire experiment.

Previous studies have reported a correlation between cognitive reports of embodiment, and proprioceptive drift[4,7,15]. However, other studies have also found that ownership does not correlate with proprioceptive drift[17,18]. In our case, we did observe a correlation between proprioceptive drift and reports of ownership (Spearman $r = 0.455, P < 0.025$), but more importantly for us, we also observed a significant correlation between the embodiment difference (average score difference of questionnaires Q1 to Q6) between the EMB and no-EMB conditions and the corresponding difference in proprioceptive drift (Spearman $r = 0.499, P = 0.013$) as shown in Fig. 3. The source data underlying Fig. 3 is provided as Supplementary Data 2.

Figure 4a shows the average distance of the ball from the robot's spherical end-effector (hand) at which a participant presses the ca-button. The source data underlying Fig. 4 is available as Supplementary Data 3. As mentioned before, the ca-button press deflected the ball away from the arm. Thus this distance also represents the minimum distance between the ball and the robot hand for that particular trial. The ball distances were observed to be $0.309 ± 0.139$ std m and $0.325 ± 0.162$ std m in the EMB and no-EMB conditions, respectively (Fig. 4a), and were not different between the two conditions $(Z(23) = 0.829, P > 0.407$, Wilcoxon signed-rank test). The participants could also largely avoid ball collisions, and the number of collisions was also not different in the EMB condition compared to the no-EMB condition $(Z(23) = 1.806, P = 0.071$, Wilcoxon signed-rank test, Fig. 4b). Overall, the left-hand performance was found to be equivalent using Kolmogorov Smirnoff tests (ball distance: $P = 0.6216$; collisions: $P = 0.6216$) in the two conditions. The overall task score was also observed to be similar between EMB and no-EMB conditions $(P = 0.6222, t = -0.4960$, T-test, $P = 0.8608$ in Kolmogorov Smirnoff tests).

Next, we examined the main task performance by analyzing the average right-hand reaction time. We expected the main task performance to be affected by the left collision avoidance task. However, the collision avoidance task required ca-button presses only every 5–8 s (compared to the right-hand button press every

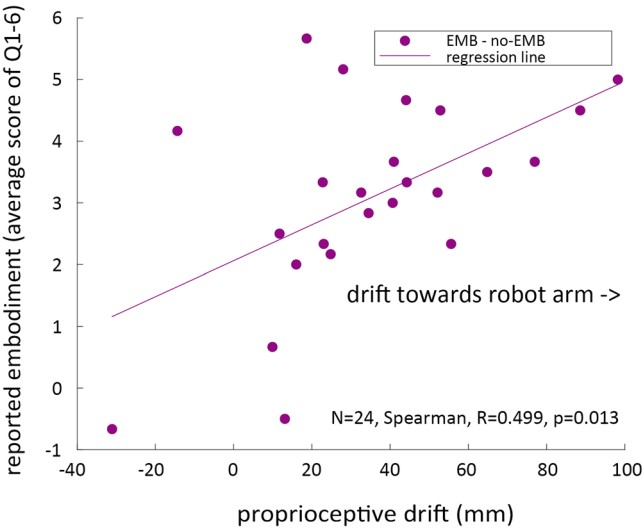

**Fig. 3 Change in embodiment correlated with change in proprioceptive drift.** Correlation of average reported score difference of questions Q1 to Q6, and difference of proprioceptive drift, between EMB and no-EMB conditions (Spearman $r = 0.499$, $P = 0.013$).

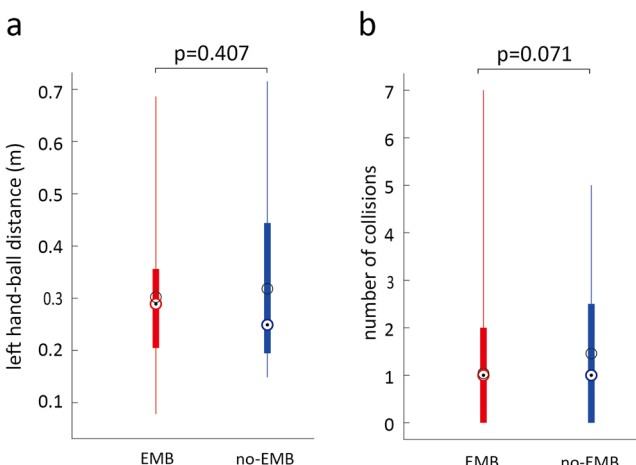

**Fig. 4 Secondary (collision avoidance) task performance.** No differences were observed between either (**a**) the distance of the ball from the left hand ($Z(23) = 0.8286$, $P = 0.407$, Wilcoxon signed-rank test), or (**b**) the number of collisions ($Z(23) = 1.806$, $P = 0.071$, Wilcoxon signed-rank test) between the EMB (red) and no-EMB (blue) conditions. This was verified with a Kolmogorov Smirnoff test for both the ball distance ($P = 0.6216$) and the number of collisions ($P = 0.6216$). The black circles show the mean and the colored circle with the dot shows the median scores across participants. The box edges show the 25th and 75th percentile of the data and the whiskers show the data range.

1–3 s) and hence the reaction times of right-hand button presses may have been modulated by its temporal proximity to a ca-button press by the left hand. To evaluate this possibility, we assimilated the reaction times from the right-hand button presses into time bins aligned with the ca-button presses (see Fig. 5a). Specifically, we collected the reaction times observed for trials within five time bins—those recorded before 1.5 s of a ca-button press (marked as [<−1500]); those recorded between 1.5 and 0.5 s before a ca-button press ([−1500, −500]); those recorded between −0.5 and +0.5 s of a ca-button press ([−500, 500]); those recorded between 1.5 and 0.5 s after a ca-button press ([500,

1500]); and those recorded after 1.5 s (and before 1.5 s of the next ca-button press) of a ca-button press ([>1500]).

Some data groups were observed to be not normal. Hence, we used an aligned rank transform[19] before performing a two-way repeated measures ANOVA on the reaction times (with the five time bins as factors) and conditions (with EMB and non-EMB conditions as factors). The two-way ANOVA showed a significant effect of time bin ($F(4,92) = 20.297$, $P < 10^{-13}$, $\eta^2 = 0.281$) as well as condition ($F(1,23) = 8.061$, $P < 0.005$, $\eta^2 = 0.037$), with no interaction ($F(4,92) = 0.812$, $P = 0.518$, $\eta^2 = 0.015$). The source data underlying Fig. 5 is available as Supplementary Data 4.

The right-hand reaction time was affected by the temporal proximity of the left-hand ca-button press. The reaction times were higher when the ca-button press was performed within the [−500, 500] ms time bin, compared to any other time bin ([<−1500]: $Z(23) = 4.114$, $P < 0.0001$; [−1500, −500]: $Z(23) = 3.285$, $P < 0.001$; [500, 1500]: $Z(23) = 4.228$, $P < 0.0001$; [1500>]: $Z(23) = 3.914$, $P < 0.0001$, post hoc Wilcoxon signed-rank test).

Crucially, the clear effect of the condition seen from the ANOVA showed that the participants could react faster with their right hand when they perceived their left robot arm to be embodied (the EMB condition) compared to when they did not (no-EMB condition).

## Discussion

In this study, we investigated how the sense of embodiment affects the attention assigned to limbs during a dual task, and consequently how this affects the main ask performance. To evaluate this issue, we developed an experimental dual task (Fig. 1) motivated by collision avoidance instances that we experience regularly in daily life. Our task required participants to perform a task requiring heavy attention with their right hand, while avoiding collisions of their left robot arm. We modulated the embodiment perceived towards the robot arm (Fig. 2), and investigated how this affected the task performance by each hand.

Note that here we were specifically interested in attention distribution in dual tasks, and its effect on the main task. We do not compare the effects of embodiment on dual-task performance with single-task performance. In fact, single-task performance has been previously shown to improve with embodiment[20]. In our experiment, we observed that the embodiment of the left robot arm enabled the participants to significantly improve their main task performance with the right hand (Fig. 5). This result has several important implications. Primarily, our results suggest that the embodiment of the robot arm modifies the attention allotment to the two arms. We observed that, while the right reaction times exhibited a general increase in the no-EMB condition (Fig. 5), the reaction time profile did not change between the conditions (see ANOVA result that shows a main effect of condition but no interaction). This suggests the attention re-allotment did not vary with time, but further studies are required to confirm this issue. Crucially, the consistent lower reaction times in the main task in the EMB condition suggests that the embodiment of the left hand enabled the participants to allot more attention to their right hand/arm. But does it mean there was a lack of attention on the left hand/arm? At least this does not seem to be so from our data, which showed that participants were able to maintain the same task performance on the left hand (see Fig. 4). These results suggest that our brain is able to better optimize the attention allotment in a dual task when the involved limbs are perceived to be part of one's body (that is, they are embodied). However, again further studies are required to clarify the exact nature of this "optimization" which enabled better performance in the main task.

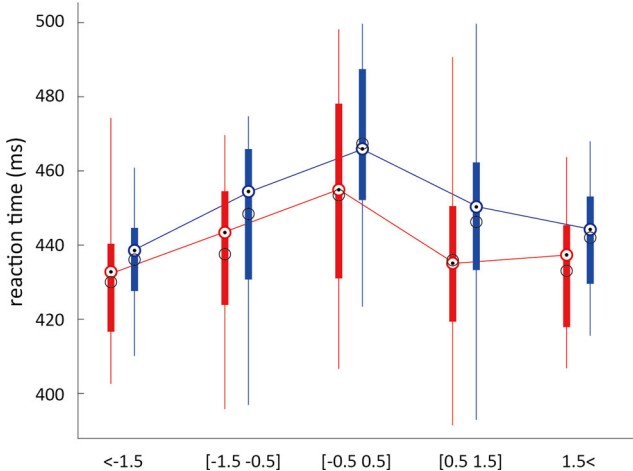

**Fig. 5 Embodiment improves main (right-hand) task performance.** The reaction times for the right-hand button press were collected from the EMB (red data) and no-EMB (blue data) conditions into time bins aligned to the left-hand ca-button presses. A two-way ANOVA showed a significant main effect of conditions ($F(1,23) = 8.061$, $P < 0.005$, $\eta^2 = 0.037$), and main effect of time bin ($F(4,92) = 20.297$, $P < 10^{-13}$, $\eta^2 = 0.281$). Embodiment of the left arm improved performance by the right hand. The black circles show the mean and the colored circle with the dot shows the median scores across participants. The box edges show the 25th and 75th percentile of the data and the whiskers show the data range.

Moreover, our results highlight attention modulation as a key effect of embodiment. Previous studies have shown that embodiment of a limb leads to increased physiological responses to perceived dangers to the limb[9,10,21,22] and increased sensitivity to sensory stimulations[11,23,24], compared to when the same limb is not embodied. Embodiment has also been suggested to improve the control of limbs[20]. Apart from this, embodiment of tools is known to change our body representations[25–27], and has also been suggested to be a key reason enabling human tool use[28–30]. Mechanisms of attention allotment, due to the embodiment (here we show with a limb, but maybe also with a tool), may provide a unified explanation for these previous results.

Finally, in regard to real life scenarios, our results suggest that task performance is indeed improved when the secondary collision avoidance is performed for one's own (embodied) arm, rather than a non-embodied object. Actual, as well as perceived, dual-task performance was better in the EMB compared to the no-EMB condition (see Q9 in Fig. 2). This result is crucial for robots used for human functional augmentation and prosthetics[31–35], as well as teleoperation[36,37], and suggests that the embodiment of these robots can enable better multi-task control and performance by the human user.

## Methods

**Participants.** In total, 30 participants took part in this study (mean age = 25.5, SD = 3.92, 23 males). Of these, six subjects who were first-time VR users were removed after we observed that first-time users were distracted and concentrated more on the left robot and task, rather than the main right-hand task. The study was approved by the local ethics committee at the University of Montpellier, France as well as the ethics committee at Waseda University, Tokyo (13 participants were appended with additional experiments in Tokyo). We used a student email mailing list to advertise the requirement of participants, and the participants were chosen on a first come first serve basis. The experiment was in Virtual reality and the data was recorded directly by the computer during the experiments. Only the experimenter and participant were present during the experiment. The experimenter was not double-blind to the conditions. All participants gave informed consent for their participation in the study. Overall, the participant number of 24 corresponded to our power analysis in G*Power 3[38] to provide 95% statistical power to achieve a medium effect size (d = 0.80) using a Wilcoxon signed-rank test against an alpha of 0.05, and to provide 80% statistical power to

achieve an effect size $f = 0.2$ in the main effect across conditions using an ANOVA repeated measures against an alpha of 0.05.

**Setup and apparatus.** The initial experiments were conducted in France. Later we had to redevelop the experiments in Japan as the initial participant numbers were deemed insufficient. The setup of the experiment was the same between the two laboratories, except for a difference in the haptic feedback device used (due to the difference in the laboratory's facility) for haptic feedbakc to the participant's left hand. This is detailed later. The data from the two participant groups were combined for analysis after checking that there was no difference between the main task (right hand) reaction times ($U = 58$, $Z = -0.753$, $P = 0.453$, Mann–Whitney $U$ test) or the left-hand collisions ($U = 67.5$, $Z = 1.778$, $P = 0.075$, Mann–Whitney $U$ test) between the two groups. The tasks are explained in the subsequent sections.

The experimental environment was constructed in virtual reality (VR). The VR space was constructed using the Unity engine (https://unity.com/ja) at a frame rate of 65 Hz. We used the VIVE VR system (https://www.vive.com/jp/) for the VR experience. The participants sat on a chair in front of a table, wore a VIVE headset during the experiment, and held the handle of VIRTUOSE haptic device (https://www.haption.com/fr/products-fr/virtuose-3d-fr.html) in their left hand. In the experiment environment in Japan, a VIVE controller was used instead of the VIRTUOSE haptic device. They rested their right hand on a keyboard on the table (see Fig. 1a).

Corresponding to the real environment, in the virtual environment as well, the participants could observe a table in front of them. They observed a keyboard and screen in front of their right hand. A virtual right hand was seen resting on the keyboard (again like in real life). In place of their left arm, they observed a robot arm connected to their body (see Fig. 1a, the upper panel). The robot arm was oriented to correspond to the left-hand configuration of the sitting participant. The robot end-effector (hand) was however linearly displaced by 10 cm from the real left hand position, towards the body midline. This was required to measure the proprioceptive drift (detailed in the section below).

When the subject moved the VIRTUOSE handle or VIVE controller, the position information was transmitted to Unity via ROS (https://www.ros.org/), and used to move the robotic left arm in the VR environment such that participants felt as if they were moving their robotic left hand.

**Task and procedure.** Figure 1b shows the experiment flow. All participants participated in two conditions- the embodiment (EMB) condition and no-embodiment (no-EMB) condition. Each condition was divided into six phases—the initialization phase, proprioceptive drift measure phase (initial), embodiment induction phase, proprioceptive drift measure phase (final) and the dual-task phase. This was followed by the questionnaire phase in which they answered twelve questions on a Likert scale. The phases are detailed below.

*Initialization phase.* The subjects adjusted their real sitting position and posture to a position where their arm coincided with the position of the arms in VR. After this, the screen blacked out for 10 s. When the VR image reappeared, the robot arm was shifted 10 cm towards the participant's body midline. The participants were not informed about this drift.

*Proprioceptive drift measure phase (initial).* The phase started with a blackout of the participant's vision in the VR. The participants were then asked to release the handle of the haptic interface and place their left hand on the table with their palm down. A flat plate (with adjustable legs) was then placed as a cover over the left hand. The plate was placed as close as possible to the hand's top surface without touching the hand. The participants were then asked to hold a pen in their right hand, and point to the index finger of their left hand by placing the tip of the pen on the cover plate. After the pointing was performed, the experimenter moved the right hand of the participant to a random location before he/she made the pointing movement again. This was done five times. We recorded the average coordinates of the pointed locations and compared them with the real position of the participant's index finger along the frontal plane, to define the initial proprioceptive drift.

*Embodiment induction phase.* A movement task followed by visuohaptic stimulation (with a paintbrush)[39,40] were used to induce a sense of embodiment in the participants toward the left robot arm and hand. Both the induction methods were utilized for the embodiment induction in all participants.

Movement task: A pink cylindrical object appears near the participant's left hand and the participant was asked to move his or her left arm to try to touch it. In the EMB condition, the movement of the robot arm in the VR was synchronized with the participant's real hand, and moved exactly like the participant's actual arm. In the no-EMB condition, the robot arm started moving after the participant's hand (delayed by 0.5–1 s) and randomly reached an object other than the one reached by the participant. A small force feedback (when VIRTUOSE was used) or small vibration feedback (when VIVE controller was used) was provided by the haptic interface when the cylinder was touched. The movement task was performed for 5 min during which 30 cylinders were presented at random locations for 7–12 s (chosen randomly), after which they disappeared even if the participant did not manage to touch it.

Visuohaptic stimulation: The participants were asked to rest their hands on the table and look at their left robot arm in VR. Their real left arm was brushed around the wrist and back of the hand by the experimenter using a paintbrush connected to a VIVE tracker. The tracker enabled us to synchronize the real brush with a brush in VR that the participants saw brushing their robot hand (or end-effector) in VR. In the EMB condition, the real and VR brushes were synchronized so that the participants felt synchronous visuohaptic stimulation. In the no-EMB condition, the VIVE tracker was detached from the real brush and the experimenter moved a VIVE tracker and the real brush independently, such that there was no synchrony between the observed movement and the felt haptic sensation. The visuohaptic stimulation was performed for 5 min.

The movement task was followed by the visuohaptic stimulation for all participants with a short break of 30 s in between.

We chose to perform both a movement task (visuoproprioceptive stimulation) and a visuohaptic stimulation due to two reasons. First, we hypothesized that using both stimulation types would increase/quicken the embodiment of the robot arm, and we also hoped that this would enable it to be maintained longer after the end of the stimulation. And second, we wanted to make participants aware (and experience) that the VR arm can be moved in synchrony with their own arm. This was required because we hoped to quantify the embodiment and/or attention during the experiment by utilizing the motor reactions by participants (such as a twitch or vibration) to the colliding ball approaching their left hand. In reality though, such a reaction was not observed in this experiment, so it is not described in the paper.

*Proprioceptive drift measure phase (final).* The final proprioceptive drift measure was calculated exactly like the initial measure. The difference between the final and initial proprioceptive measures provided us with the proprioceptive drift induced after EMB or no-EMB for each participant.

*Dual-task phase.* The participants worked on the experiment dual task in this phase. The dual task required them to perform a *main task* with their right hand, and a *secondary* collision avoidance task with their left hand.

The main task required the participants to watch two rectangular panels presented in the right visual field of the VR environment, in front of their right hand. The panels changed their colors (red, blue, or yellow) randomly every 500 ms. The participants were instructed to "press the space key as soon as the colors of the two rectangles became the same", which happened between one to 2.5 s. Overall the participants were presented with 48 same color stimuli (which required a keyboard press) per session. A correct press earned the participants 20 points. The participants were penalized −10 points when they missed pressing a button when the panel colors were same, or pressed the keyboard when the panel colors did not match. We recorded the right-hand reaction times as measures of the main task performance by each participant.

The participants were presented with a pink ball (5 cm in diameter) in their left visual field. The ball flew near the left robot hand/arm of the participants in a pseudo-random trajectory within $1 \times 0.8 \times 1.5$ m in the virtual environment around the left hand of the participant. The ball approached the robot hand/arm every 5–8 s. The ball approached the hand following one of seven manually designed trajectories in each collision. Each of the trajectories was designed by choosing 6 via points to the participant's hand position and back. A trajectory generator provided in the UNITY software was used to develop the trajectory through these via points given the velocities at the via points (which were set between 0.25 m/s and 0.75 m/s at each point).

As the secondary task, the participants were instructed to prevent the ball from hitting their left hand or arm. They were instructed to press the collision avoidance button (ca-button), under their left thumb on the handle of the haptic device, whenever they felt that the ball may collide with the left hand or arm. Pressing the ca-button resulted in the ball being deflected the ball away from their arm. Collisions resulted in penalization of five points. The participants were however rewarded 1 point if they could press the ca-button after the ball was closer than 30 cm to their hand. Any presses when the ball was beyond 30 cm earned them no points. This scenario enabled us to quantify the secondary task performance by the distance between the hand and the ball when the ca-button was pressed, and the collisions incurred by the participants.

The dual-task phase consisted of three 2-min trials. The participants performed the above-mentioned tasks with their two hands in every trial. The dual task was the same in both the EMB and no-EMB conditions. We utilized the data from all trials except the first 30 s data of the first trial, in which the performance was assumed to not have stabilized.

Finally, at the end of each condition, each participant answered the following 12 questions on a seven-point Likert scale.

From 1 (not at all) to 7 (very strongly), it seems like…

1. The robot arm is part of your body
2. The robot arm is your arm
3. The robot arm belongs to you
4. The robot arm is in the location where your arm is
5. You could push an object with the arm you see
6. You could move the arm you see

7. You could perform the left-hand task well
8. You could perform the right-hand task well
9. You could perform tasks on each arm equally well
10. The task on the right hand disturbed the task on the left hand
11. You were anxious about your left-hand task
12. You were anxious about your right-hand task.

The first three questions estimated the ownership perceived towards the left robot hand, by a participant. The fourth question estimates the perceived location of the robot arm, while questions five and six estimated the sense of agency perceived towards the left robot hand by a participant. The average score by a participant across questions one to six was taken as a measure of embodiment perceived towards the robot arm[23].

Questions seven to ten estimated the participant's perception of their performance and were asked to verify if the participants cognitively felt differences in performance between the two conditions. Given the relative ease of the left-hand task, we did not apriori expect a change in anxiety between the conditions and questions eleven and twelve served as control questions in the questionnaires.

**Statistics and reproducibility.** All data groups were first checked for normality using the Shapiro–Wilk test. Data groups which were found to be normal ($P > 0.05$) were treated using parametric tests, namely the T-test (embodiment drift in Fig. 2). Data groups that were found to be non-normal were compared using the Wilcoxon sign-rank test (questionnaire score in Fig. 2, ball distance and the number of collisions in Fig. 4) and using the Aligned Rank Test before an ANOVA (reaction time in Fig. 5) and analyzed using Spearman correlation (Fig. 3).

**Reporting summary.** Further information on research design is available in the Nature Research Reporting Summary linked to this article.

## Data availability
The final dataset used for the plots is provided as supplementary materials. Detailed datasets generated during and/or analyzed during the current study are available from the corresponding author on reasonable request.

## Code availability
The codes generated during and/or analyzed during this study are available from the corresponding author on reasonable request.

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

## Acknowledgements
This work was supported by the JST ERATO Grant Number JPMJER1701. The authors thank Guillaume Gourmelen and Adrien Verhulst for their help with the VR and robot setup.

## Author contributions
Conceptualization: Y.I. and G.G.; methodology: Y.I. and G.G.; setup development: Y.I. and B.N.; data collection: Y.I.; data analysis: Y.I. and G.G.; writing—original draft: Y.I. and G.G.; writing—review & editing: Y.I., B.N., H.I., and G.G.; funding acquisition, H.I. and G.G.; resources: Y.I. and G.G.; supervision, B.N., H.I., and G.G.

## Competing interests
The authors declare no competing interests.
