## [Peer Review file · Communications Biology]

Reviewers' comments:

Reviewer #1 (Remarks to the Author):

The study's topic is embodiment, the sense that the effector we use to perform an action, such as our hand when reaching for an object, is our own. Embodiment is a very interesting topic in neuroscience, as it is relevant to sensorimotor control and the experience of behaving, at a minimum, and may be relevant to questions of consciousness and agency that might be relevant to decision making.

The authors asked whether a sense of embodiment facilitates the performance of a dual task. The more general idea of interest is whether embodiment enhances our ability to perform complex motor tasks. Participants were asked to perform two tasks simultaneously: press a key with their right hand when the color of a square changed (primary task) and move their left arm and hand out of the way of (virtual) balls that moved toward their left hand (secondary task). The outcome measure was the reaction time of the right hand's button presses: how fast did participants respond to a color change when simultaneously performing the secondary task with a robot arm that felt like their own (embodiment condition) vs. not their own (no-emb condition). Reaction times were shorter in the embodiment condition, suggesting that perceiving a tool as part of one's own body frees up resources, presumably attentional resources, to perform another task faster.

The question asked in this study is interesting and important. The study was very well designed and executed, and the data was analyzed appropriately. The result is of considerable relevance to our understanding of sensorimotor control and of the computational mental processes that govern actions. The paper is well written and easy to read.

One quality of the study is that the authors demonstrated that the secondary task was not affected by the embodiment state. Participants were able to avoid collisions by using their left hand to guide a robot arm that either felt like their own or did not, and performed equally well in those conditions. A difference in performance could have been a major confounder and the authors did well to check that there was no difference (Fig. 3).

Another quality is that the authors measured the sense of embodiment that participants achieved, and demonstrated that there was a measurable difference between the embodiment and no-emb conditions (Fig. 2).

A very interesting aspect of the results was the time-sensitivity of the RT reduction: it was greatest when the two actions occurred nearly simultaneously, and dropped to nearly 0 when the right and left hand actions were separated by more than 1.5 seconds (Fig. 4). This time sensitivity is also exhibited by the right hand's RT. This congruence makes it easier to believe that embodiment is improving control of a competing task rather than improving sensorimotor performance non-specifically. However, this is an impression conveyed by the graph. The lack of significant interaction reported in line 191 indicates that this result is only suggested by the graph but is not statistically significant.

I have the following concerns.

1. It would be worthwhile to explicitly comment on the scale of the magnitude of the effect reported. By including time separations greater than 1.5 s between R and L hand tasks, the authors effectively included a "control" condition equivalent to single-task performance. The difference in RT between the first red data point in Fig. 4a and the third (middle) one, about 30 ms, is an indication of the RT cost of the secondary task on the first task. The greatest amount of RT reduction was around 25 ms (center point in Fig. 4a), which suggests that the benefit of embodiment is of the same order of magnitude as the cost of dual tasking.

2. Line 206. The correlation between one measure of embodiment and change in RT is a question of

great interest because a correlation would strengthen the cause-effect conclusion about embodiment on RT saving. However, I disagree with calling this correlation "strong", as the authors do. The correlation coefficient is 0.5, which corresponds to a p value of 0.057. I would refer to this as a possible correlation. The graph makes a stronger case for a correlation, as it suggests an inverse correlation that is even stronger than linear, i.e. the "outlier" data points actually suggest a strong relationship (even though they introduce noise that weakens the correlation coefficient).

Minor suggestions:

Line 14. I would strike out "also" as it seems redundant to me.

Line 16: I suggest striking out the word "embodied" here and not mention it in the abstract unless you can also define it here. I think it's ok to wait to introduce it until the Introduction where you can properly define it.

Line 193: change "form" to "from"

Line 215: remove the comma after "attention".

The references are cited by author names but are listed by number in order of appearance. This makes it very difficult and frustrating to find references for given citations in the text. I suggest the authors review the journal's guidelines for references.

Reviewer #2 (Remarks to the Author):

The authors used a virtual reality experiment to examine how embodiment might influence attentional distribution across the hands. They tested participant performance in two simultaneous tasks, one of which was performed by the right hand, the other by the left hand, which could be represented by an embodied virtual hand or an unembodied virtual hand. They propose that embodiment of the left hand improved task performance with the right hand. The authors conclude that embodiment of one limb can improve attentional allocation to the other, potentially providing an explanation of our ability to maintain focus on the world without, for example, bumping into things.

I really like the idea of this experiment, and I think it aims to answer some important unanswered questions. However, there are several unusual methodological choices that I think could be improved upon in a follow up experiment before strong conclusions are made. In addition, the authors seem to throw out 2/3 of their data, in a way that seems to be inappropriate, making it hard to have confidence in their conclusions.

Introduction:

1. Line 36: Why should visual attention be of specific importance here? We don't need to look at our arm to avoid it bumping into things. Surely attention can be considered multisensory in the scenarios the authors are describing, with proprioception being of particular importance?

2. Lines 45-47: I think the authors need to expand on the hypotheses mentioned here. I do not immediately follow the logic behind them. A more coherent explanation of what they mean by 'attention' and 'awareness' would be useful.

Method:

3. Line 377: What reason did the authors have to expect an effect size of $d = 0.75$?
4. Line 411: Did the authors check if participants noticed this shift of the robot arm?
5. Line 415: I am a bit wary of the proposal that proprioceptive drift captures body ownership. See for example Rohde et al. (2011; PLoS One) and Abdulkarim & Ehrsson (2016; Attention, Perception, & Psychophysics)
6. Line 444: Why did the participants rest their hand for visuo-haptic stimulation? Surely altering the distance between the real hand and the perceived robot hand in VR would invalidate the previous movement task induction? In general I don't understand why two illusion induction methods were used (most body ownership experiments don't require this), and why a 30 second break was given between them. The movement task appears to be sufficient.
7. Line 456: Why did the authors choose to take the second proprioceptive drift measure before the actual task? Are they certain that the drift would have held throughout the main experiment? How can they be certain that drift did not occur also in the no-EMB condition? Visual information about limb position can have strong effects on proprioception (Holmes et al., 2006; Perception & Psychophysics).
8. Line 462: Did they also ensure that the right virtual hand was 'embodied'? Did it press a key when the participant did?
9. Line 467: Please provide details on how many stimuli were presented
10. Line 468: I suggest that the authors avoid the phrase 'on the other hand', given the experiment they are running!
11. Line 470: Were only correct reaction times analysed? Also, whilst the arbitrary scores are appropriate for helping participant engagement, I don't understand why they should be analysed in this way. Why should a correct press be worth 20 and an incorrect press be worth -10? For the purpose of analysis, surely the most obvious approach would be (correct presses) - (incorrect presses)? In addition, looking at Figure 4b suggests that some participants did very poorly in this task. Why was this?
12. Line 487-489: Was there any benefit to the participants for gaining more points? Why did the authors decide not to award points when the ball was greater than 30 cm away? Surely heightened awareness would promote an earlier response?
13. Line 507-508: I'm not sure I follow the logic here. The authors propose that their embodiment manipulation worked only in the first trial, since they only observe significant differences for their task in that time. But that is not their measure of embodiment (questionnaires/drift). If they have no evidence from these measures to compare the different trials, it is not right to make such inferences. Even if differences in embodiment might alter behavioural outcomes, it doesn't mean that if there is no difference in behavioural outcomes, embodiment has not occurred. In addition, their questionnaires were taken at the end of all the trials. You might expect no evidence for embodiment to show itself here, if 2/3 trials were not experienced as embodied as the authors suggest. This reverse inference unfortunately comes across like cherry-picking, and quite possibly invalidates their conclusions. A follow-up experiment, with illusion inductions and embodiment measurements performed for each trial, as well as addressing the other queries I raise in my review, would help validate the findings.
14. Lines 536-538: What is the purpose of these questions?

Results:

15: Figure 2: Doesn't 'ownership' also include question 3? The authors also appear to mix up ownership, agency, and location in lines 109-110.

16. Lines 165-171: It is not correct to claim evidence for 'no difference' based on non-significant p-values. Bayesian analysis or equivalence testing would be appropriate.

17. Lines 184-196: Was this analysis post hoc? If so, a follow up study with the left and right hand stimuli arriving near or far in time from each other would help validate the finding.

18. Line 205: I'm not sure why the 10 minutes is interesting. The rubber hand illusion occurs after less than 30 seconds of induction!

19. Line 211: This correlation is not statistically significant, and should not be interpreted as such. In general, one should be very sceptical of correlations run on less than 30 participants. See this blog post for some reasons why: <https://garstats.wordpress.com/2018/06/01/smallincorr/>

Discussion:

20. Lines 230-233: Again, it is not appropriate to draw conclusions based on non-significant p-values.

21. Lines 236-237: It would be nice to have some more detailed discussion of this idea. Is attending to an embodied limb 'easier' or 'more efficient', therefore leaving more attentional resources for other tasks?

22. Lines 237-239: How does this account for the virtual right hand? Do the authors think this hand was also embodied?

23. Line 251-252: This is an incredibly strong statement and I do not believe the authors have adequate justification for it.

24. Lines 2256-257: Do their results actually provide evidence in favour of an argument for 'hand held objects'? I'm not sure how this claim logically follows from their experiment. In both conditions an entire robot arm and hand was seen, not a hand held object. I also don't see how their claim made in lines 260-261 follow, for similar reasons. The robot hand was not a hand held object, so they are making an inference here that is not backed up by their data.

Reviewer #3 (Remarks to the Author):

Iwasaki et al studied the effect on embodiment on attention allocation in dual tasking. They employed a virtual rubber hand illusion, using asynchronous stimulation as control, and then exposed subjects to an 'easy' collision avoidance task with their left hand, and a 'difficult' cued button-press task with their right hand. There was no illusion vs non-illusion difference in task performance with respect to the left-hand collision avoidance task, but the illusion was associated with increased task score ($p=0.047$), decreased RT ($p=0.037$), and a correlation between RT change and proprioceptive drift change ($p=0.057$) for the right-hand cued button press task. The authors conclude that their results "suggest that embodiment of a limb improves attention allotment for dual task performance with it."

Although I find the idea and rationale for this study interesting, there are serious weaknesses in experimental design and the conclusions are based on just barely significant results, which make me unable to recommend this study for publication. Below are my major points.

- It is impossible to determine that the increase in cued-button-press task performance has anything to do with the employment of a dual task, because there is no single-task control condition. It could be the case that the observed increases in cued-button-press task performance is simply a general effect of experiencing the rubber hand illusion. In order to make inferences about the effect of embodiment on attention allocation during a dual task, the authors must conduct a control experiment with a single task (i.e., the exact same experiment but without the left-hand collision avoidance task). If the task performance effect disappears in the single-task experiment, and there is an interaction between (single-task vs dual-task) and (embodiment vs non-embodiment), one may be able to make meaningful inferences about the effect on embodiment on attention allocation during a dual task.

- The results are statistically weak. The effect on task score ($p=0.047$) and RT ($p=0.037$) are just barely $p<0.05$ significant, and the correlation between RT change and proprioceptive drift change is actually not ($p=0.057$). Furthermore, the correlation seems to be driven by 1 outlier showing 70 ms RT change and -40 mm drift. And the task score difference (Fig 2B) appears to be dependent on excluding one outlier (if interpret the Fig 2B correctly).

- The weak results may be due to the sample size of 16, which is very small. The authors' power calculation aimed to "provide 80% statistical power to achieve a medium effect size ($d = 0.75$)". However, $d=0.75$ corresponds to a large effect (traditionally, $d=0.5$ is considered "medium", and $d=0.8$ is considered "large"). What basis do the authors have for assuming that the effect size should be this large?

Questionnaires:

- Why are there no control questions, to control for suggestibility and task compliance?
- Fig 2 states that Q3 ("The robot arm belongs to you") is a "location" statement. However, this statement is clearly an ownership statement and should be analyzed as that.
- The results section: "We observed that the ownership (average score in Q1 and Q2), agency (average of Q3 and Q4) as well the sense of location (average of Q5 and Q6) towards the robot arm was [...]". There is some serious confusion about the numbering of the statements here. In the methods section, agency statements are Q5 & Q6, not Q3 & Q4. Also, as stated in the previous point, Q3 ("The robot arm belongs to you") cannot be regarded and analyzed as a "location" statement - this is an ownership statement.

- Throughout the manuscript, the authors use the grocery shop basket as an example of everyday life dual tasking. They state: "Ownership of a limb is known to improve its visual awareness (Hoort, Reingardt, and Ehrsson, 2017), suggesting that we will be more aware of obstacles to our swinging arm, than to the swinging basket in our arms, which we do not feel a sense of embodiment towards." This notion is not necessarily true. A basket held in one's hand should be considered as tool, which, in many respects, have been shown to be incorporated into the body image (even though people don't have ownership feelings over tools). The study by Hoort et al compared the visual awareness of an owned rubber hand versus an un-owned rubber hand, which has no implications for the awareness of one's own body versus a tool (because an 'un-owned' rubber hand is not a tool.)

- The language throughout the paper needs to be improved. I would recommend a professional language service.

Minor:

- Abstract: line 21. "don't" -> "did not"
- Line 141. "let" -> left.
- Please define what a "ca-button" is.
- Line 160. "cm" -> "m"
- Line 190. " $p < 10^{-8}$ " - A number is lacking.
- Throughout the manuscript, exact p values are often described as " $p < 0.XX$ ", when it should say

"p=0.XX".

We thank the reviewers and editor for their time and constructive comments that have contributed to greatly improve the manuscript. Below we provide our answers for each of the reviewer comment. The corresponding major changes in the manuscript have been highlighted in the “changes_underlined_manuscript.docx” file.

We would like to apologize for the delay in the resubmission of the manuscript. The first author moved from the lab in France, where the initial experiments were conducted, to Japan at the start of the COVID pandemic. And in order to increase the participant numbers in the study (which was one of the key requests from the reviewers), we had to re-develop the experiments in the new lab, and apply and acquire the ethical approval, and recruit additional participants, all of which was slow during the COVID restrictions.

The main changes in the manuscript are the follows

- 1) As per the comments by a reviewer, we have increased the number of the participants and collected data from 13 additional participants. The data from the additional participants was combined with the previous participants after checking that there were no difference between the main task (right hand) reaction times ($U=58$, $Z=-0.753$, $p=0.453$, Mann-Whitney U test) or the left hand collisions ($U=67.5$, $Z=1.778$, $p=0.075$, Mann-Whitney U test) between the two groups. In total, we report results from 24 participants here. Some participants without any previous VR experience were omitted from the analysis.
- 2) We have reformulated the introduction to better introduce the motivation as per the recommendation of the reviewers.
- 3) Several changes in the description of the experiment and in the discussion as per reviewer comments.

Reviewer #1 (Remarks to the Author):

R: The study's topic is embodiment, the sense that the effector we use to perform an action, such as our hand when reaching for an object, is our own. Embodiment is a very interesting topic in neuroscience, as it is relevant to sensorimotor control and the experience of behaving, at a minimum, and may be relevant to questions of consciousness and agency that might be relevant to decision making.

The authors asked whether a sense of embodiment facilitates the performance of a dual task. The more general idea of interest is whether embodiment enhances our ability to perform complex motor tasks. Participants were asked to perform two tasks simultaneously: press a key with their right hand when the color of a square changed (primary task) and move their left arm and hand out of the way of (virtual) balls that moved toward their left hand (secondary task). The outcome

measure was the reaction time of the right hand's button presses: how fast did participants respond to a color change when simultaneously performing the secondary task with a robot arm that felt like their own (embodiment condition) vs. not their own (no-emb condition). Reaction times were shorter in the embodiment condition, suggesting that perceiving a tool as part of one's own body frees up resources, presumably attentional resources, to perform another task faster.

The question asked in this study is interesting and important. The study was very well designed and executed, and the data was analyzed appropriately. The result is of considerable relevance to our understanding of sensorimotor control and of the computational mental processes that govern actions. The paper is well written and easy to read.

One quality of the study is that the authors demonstrated that the secondary task was not affected by the embodiment state. Participants were able to avoid collisions by using their left hand to guide a robot arm that either felt like their own or did not, and performed equally well in those conditions. A difference in performance could have been a major confounder and the authors did well to check that there was no difference (Fig. 3).

Another quality is that the authors measured the sense of embodiment that participants achieved, and demonstrated that there was a measurable difference between the embodiment and no-emb conditions (Fig. 2).

A very interesting aspect of the results was the time-sensitivity of the RT reduction: it was greatest when the two actions occurred nearly simultaneously, and dropped to nearly 0 when the right and left hand actions were separated by more than 1.5 seconds (Fig. 4). This time sensitivity is also exhibited by the right hand's RT. This congruence makes it easier to believe that embodiment is improving control of a competing task rather than improving sensorimotor performance non-specifically.

A: We thank the reviewer for his positive comments.

R: However, this is an impression conveyed by the graph. The lack of significant interaction reported in line 191 indicates that this result is only suggested by the graph but is not statistically significant.

I have the following concerns.

1. It would be worthwhile to explicitly comment on the scale of the magnitude of the effect reported. By including time separations greater than 1.5 s between R and L hand tasks, the authors effectively included a "control" condition equivalent to single-task performance. The difference in RT between the first red data point in Fig. 4a and the third (middle) one, about 30 ms, is an indication of the RT cost of the secondary task on the first task. The greatest amount of RT reduction was around 25 ms (center point in Fig. 4a), which suggests that the benefit of embodiment is of the same order of magnitude as the cost of dual tasking.

A: We thank the reviewer for pointing to this issue. We did in fact design the analysis hoping to use the time bins further away from the left hand ca-button press as single trials. However, as the reviewer points out, we did not observe an interaction in our ANOVA analysis, showing that the embodiment effects did not change across time bins. We therefore refrain from discussion this issue in the manuscript.

However, please note that the purpose of this study was to specifically compare the attention distribution between two limbs given a dial task, when one limb is embodied or not, and not to compare the effects of embodiment between single and dual tasks. Task performance in a single task has been in fact previously shown to improve with embodiment (Newport, Pearce and Preston, 2009). The dual task here serves us as a tool that helps estimate the inter-limb attention distribution (through a measure of task performance by each limb).

R:2. Line 206. The correlation between one measure of embodiment and change in RT is a question of great interest because a correlation would strengthen the cause-effect conclusion about embodiment on RT saving. However, I disagree with calling this correlation "strong", as the authors do. The correlation coefficient is 0.5, which corresponds to a p value of 0.057. I would refer to this as a possible correlation. The graph makes a stronger case for a correlation, as it suggests an inverse correlation that is even stronger than linear, i.e. the "outlier" data points actually suggest a strong relationship (even though they introduce noise that weakens the correlation coefficient).

A: We agree with the reviewer. However, after the experiment was conducted with more participants, we could no longer observe a correlation between RT and the measure of embodiment. We had to therefore remove this figure from the updated manuscript.

Minor suggestions:

R: Line 14. I would strike out "also" as it seems redundant to me.

A: We thank the reviewer and have corrected this typo.

R: Line 16: I suggest striking out the word "embodied" here and not mention it in the abstract unless you can also define it here. I think it's ok to wait to introduce it until the Introduction where you can properly define it.

A: We appreciate the reviewer's concern but we feel we have to keep 'embodiment' here to clearly specify the aim and scope of the study. We also have 'embodiment' in the title for the same reason.

RLine 193: change "form" to "from"

A: We have corrected this typo.

R: Line 215: remove the comma after "attention".

A: Thank you. We have deleted the comma.

R: The references are cited by author names but are listed by number in order of appearance. This makes it very difficult and frustrating to find references for given citations in the text. I suggest the authors review the journal's guidelines for references.

A: We are sorry for this. We have now changed the references to conform to the journal format.

Reviewer #2

The authors used a virtual reality experiment to examine how embodiment might influence attentional distribution across the hands. They tested participant performance in two simultaneous tasks, one of which was performed by the right hand, the other by the left hand, which could be represented by an embodied virtual hand or an unembodied virtual hand. They propose that embodiment of the left hand improved task performance with the right hand. The authors conclude that embodiment of one limb can improve attentional allocation to the other, potentially providing an explanation of our ability to maintain focus on the world without, for example, bumping into things.

I really like the idea of this experiment, and I think it aims to answer some important unanswered questions.

A: We thank the reviewer for his/her positive comments

R: However, there are several unusual methodological choices that I think could be improved upon in a follow up experiment before strong conclusions are made. In addition, the authors seem to throw out 2/3 of their data, in a way that seems to be inappropriate, making it hard to have confidence in their conclusions.

A: In the previous version, we only considered the first session after embodiment induction so as to observe the attentional effect during the period when we expected the effects of embodiment to be strongest. However, we have now increased the number of participants in the study, and have also included all the sessions in the analysis.

R: 1. Line 36: Why should visual attention be of specific importance here? We don't need to look at our arm to avoid it bumping into things. Surely attention can be considered multisensory in the scenarios the authors are describing, with proprioception being of particular importance?

A: We agree with the reviewer that attention in different sensory modalities may enable collision avoidance. We have removed the specific reference to ‘visual’ attention in the introduction.

2. Lines 45-47: I think the authors need to expand on the hypotheses mentioned here. I do not immediately follow the logic behind them. A more coherent explanation of what they mean by ‘attention’ and ‘awareness’ would be useful.

R: ‘Awareness’ has been used in literature to refer to our perception of the spatial location of our limbs and our environment while attention refers to multisensory perception (like the reviewer mentions in the last answer) often quantified by our sensitivity to our body parts and environment using different sensory modalities. We are interested to investigate attention in this study and hence use a dual task which has been a standard tool to verify attention allotment. We refer to awareness as it is a close concept and because it has been previously shown to be modulated by ownership. We agree that mentioning these together makes the logic unclear. We have reformulated the abstract and introduction to focus more on ‘attention’.

R:3. Line 377: What reason did the authors have to expect an effect size of $d = 0.75$?

A: We had made a preliminary experiment with a separate participant group to estimate the effect size, and the corresponding participant numbers. However, we have now increased the number of participants to 24 after the request by a reviewer. This number corresponds to 95% statistical power to achieve a medium effect size ($d = 0.80$) using a Wilcoxon signed rank test ($\alpha = .05$), and 80% statistical power to achieve an effect size $f = 0.2$ ($\eta^2 = 0.03$) in the main effect across conditions using a repeated measures ANOVA. We can confirm that the effects size of the results equal or better these values for the main results supporting our conclusion- For example, the ANOVA main effects we observed to have an effect size of $\eta^2 = 0.037$ (EMB condition) and $\eta^2 = 0.281$ (time bin), and the Wilcoxon signed-rank test between the EMB and no- EMB conditions showed an effect size of $d = 1.189$ (proprioceptive drift difference), $d = 1.9187$ (questionnaire).

We have added the following statement to Line 375-378 corresponding to this

“Overall the participant number of twenty four corresponded to our power analysis in G*Power 3 to provide 95% statistical power to achieve a medium effect size ($d = 0.80$) using a Wilcoxon signed rank test against an alpha of .05, and to provide 80% statistical power to achieve an effect size $f = 0.2$ in the main effect across conditions using an ANOVA repeated measures against an alpha of .05.”

R: 4. Line 411: Did the authors check if participants noticed this shift of the robot arm?

A: After the experiment, we interviewed the subjects to see if they had noticed the shift of the robot arm. None of the subjects noticed it. We have updated this information on lines 144-146:

“Finally, the proprioceptive drift was observed to be higher in the EMB condition, compared to the no-EMB condition ($Z(23)=-3.857$, $p<0.001$), even though none of the participants reported noticing this shift when asked after the end of the entire experiment.”

R: 5. Line 415: I am a bit wary of the proposal that proprioceptive drift captures body ownership. See for example Rohde et al. (2011; PLoS One) and Abdulkarim & Ehrsson (2016; Attention, Perception, & Psychophysics)

A: It is true that some studies have found that the reports ownership alone do not correlate with proprioceptive drift. However, in our study we did observe a correlation between the reported ownership difference (averaged across Q1, Q2, Q3) between the EMB and no-EMB conditions and the corresponding difference of proprioceptive drift (Spearman $r=0.455$, $p<0.025$).

Crucially for us, we observed a significant correlation between the average score difference of questionnaires that represent overall embodiment (Q1 to Q6) between the EMB and no-EMB conditions and the corresponding difference of proprioceptive drift (Spearman $r=0.499$, $p<0.013$). This corresponds to reports by several previous studies that show that cognitive reports of embodiment (when including measures of ownership, agency and location) correlates with the proprioceptive drift (Longo, Schüür, Kammers, Tsakiris and Haggard, 2008; Botvinick and Cohen, 1998; Holmes, Full, Koditschek and Guckenheimer, 2006).

We have appended the correlation results between ownership and proprioceptive drift on line 150-154

R:6. Line 444: Why did the participants rest their hand for visuo-haptic stimulation? Surely altering the distance between the real hand and the perceived robot hand in VR would invalidate the previous movement task induction? In general I don't understand why two illusion induction methods were used (most body ownership experiments don't require this), and why a 30 second break was given between them. The movement task appears to be sufficient.

A: There seems to be a misunderstanding here. The robot arm was also stationary during the visuo-haptic stimulation session. The visuo-haptic stimulation was similar to the classic procedure used for the rubber hand illusion. An experimenter brushed the real hand of the participant with a brush while the participant observed the robot hand being brushed at the same location in VR. Hence there was no alteration in the perceived distance between the real hand/arm and the robot hand/arm at any instance.

We chose to perform both a movement task (visuo- proprioceptive stimulation) and a visuo-haptic stimulation due to two reasons. First, because we hypothesized that using both stimulation types would increase/quicken the embodiment of the robot arm, and also enable it to be maintained longer after the end of the stimulation. And second, in addition to the standard visuo-haptic

stimulation, we chose the movement task because we wanted to make participants aware (and experience) that the VR arm *can* be moved (in synchrony with their own arm). This was required because we hoped to quantify the embodiment and/or attention during the experiment by utilizing the motor reactions by participants (such as a twitch or vibration) to the colliding ball approaching their left hand. In reality though, such a reaction was not observed in this experiment, so it is not described in the paper.

We have clarified this point in lines 464-472 of the updated manuscript.

R:7. Line 456: Why did the authors choose to take the second proprioceptive drift measure before the actual task? Are they certain that the drift would have held throughout the main experiment? How can they be certain that drift did not occur also in the no-EMB condition? Visual information about limb position can have strong effects on proprioception (Holmes et al., 2006; Perception & Psychophysics).

A: There seems to be a misunderstanding on the reviewer's part here. The proprioceptive measure was performed both before and after the embodiment induction session. And the difference of the two measure defined the proprioceptive drift. Importantly, the proprioceptive drift measurement was performed in both the EMB and no-EMB conditions. The drift measurement in the no-EMB condition was used as a control for-

- a) Possible visual effects (as mentioned by the reviewer) on drift: Note that the visual feedback during embodiment induction was same for both the EMB and no-EMB conditions.
- b) Possible haptic/proprioceptive effects on drift: the haptic feedback during embodiment induction was quantitatively same for both the EMB and no-EMB conditions. In fact the only difference in between the two conditions was in the coherence of the visual and haptic (visuo-haptic stimulation) or proprioceptive (in case of the movement task) stimulation – which is the key requirement for embodiment induction.
- c) Time effects on drift: Again, the duration of the induction was same between the EMB and no-EMB conditions, and importantly, the duration of the main task *after* the induction phase was same in the two conditions.

The difference between the drifts in the EMB and no-EMB conditions in plotted in Fig. 2 (last pair of bars) and in the correlation plot of Fig. 3 and served as a behavioral estimation of the embodiment (of the robotic left arm) induced in the participants.

The question of whether the drift (and the embodiment) remained through the main experiment is a valid concern of the reviewer. It is quite possible that the embodiment (and drift) slowly diminished through the main experiment sessions. However, please note that we still see

differences between the EMB and no-EMB conditions (fig. 3), even though both followed the same protocol. This shows that some embodiment effect did remain throughout our task. In fact, as mentioned in the text, the main task was exactly the same (in terms of the feedbacks provided to the participants and actions required by them) in the EMB and no-EMB conditions, which differed only in the initial induction phase of the experiment.

R:8. Line 462: Did they also ensure that the right virtual hand was ‘embodied’? Did it press a key when the participant did?

A: The virtual finger did not move in the experiment but we do not believe this affected the embodiment of the right hand/arm, and our results. This was because, given the fact that the participant finger rested on the key board throughout the experiment, the finger movements required to press the keyboard were minimal. Furthermore, the finger movements were performed while attending to visual cues which were purposely designed to demand high visual attention (please see right hand task description) by the participants. This we believe would have limited the participants from attending to the virtual finger.

On the other hand, we believe the location of the virtual arm and the visuo-haptic feedback corresponding to the keyboard touch (that was present through the experiment), were more prominent determinants of the embodiment of the right hand. For this, we ensured that the virtual arm was aligned to have the same position as the real arm and was also shown resting on the keyboard touching the same key as the real right hand. This ensured that the visuo-haptic feedback corresponding to the keyboard touch was synchronized for the right hand.

Finally, even if we assume that there was indeed some loss of embodiment due to the absence of the virtual right finger movement, then this loss should be same in both the EMB and no-EMB conditions and therefore, not expected to contribute to our observations.

R:9. Line 467: Please provide details on how many stimuli were presented

A:Regarding the color panel on the right hand task, the display changed every 0.5 seconds, but the colors always matched at least once out of five times. This means that on average, the same color was displayed once every 2.5 seconds. There were 48 stimuli per session (120 seconds). This has been added to Line 497-500.

“The participants were instructed to “press the space key as soon as the colors of the two rectangles became the same”, which happened roughly every one to 2.5 seconds. Overall the participants were presented with 48 same color stimuli (which required a keyboard press) per session.
“

R:10: Line 468: I suggest that the authors avoid the phrase ‘on the other hand’, given the

experiment they are running!

A: We thank the reviewer for this comment. He/she is indeed right about this point. We have now replaced “on the other hand” with “however” or other equivalent terms throughout the manuscript.

R:11. Line 470: Were only correct reaction times analysed? Also, whilst the arbitrary scores are appropriate for helping participant engagement, I don't understand why they should be analysed in this way. Why should a correct press be worth 20 and an incorrect press be worth -10? For the purpose of analysis, surely the most obvious approach would be (correct presses) – (incorrect presses)? In addition, looking at Figure 4b suggests that some participants did very poorly in this task. Why was this?

A: The color cues for the right hand changed every 500 ms, and the participants were required to press the keyboard key when the color cues were same. We included all presses that happened within 500 ms of presentation of a cue. ‘Incorrect’ presses are defined as presses that were made when the two color cues were not same. We do not include these presses in our analysis.

The difference between the correct and incorrect presses however, did not change across the two conditions ($Z(23)=-1.114$, $p=0.267$, Wilcoxon signed-rank test).

Regarding the scoring rule, the primary purpose of the chosen numbers was to ensure the participants are not too demotivated after wrong presses. But after the inclusion of the new participants, we no longer observe a significant difference between the task scores in the two conditions. We therefore, no longer plot the scores.

The inter-participant performance was not large. Please see the boxplot edges which show the 25th and 75th percentile in the previous draft (Note that the whiskers show the data range, and hence are susceptible to outliers). We have however removed this figure in the updated version as we do not see a difference in the scores between the two conditions.

R:12. Line 487-489: Was there any benefit to the participants for gaining more points? Why did the authors decide not to award points when the ball was greater than 30 cm away? Surely heightened awareness would promote an earlier response?

A: A strategy that makes this game easier is to keep avoiding collisions, regardless of the ball's position, by constantly pressing the ca-button in rapid succession, without paying particular attention to the ball's movement. To discourage participants from taking this strategy, we needed a game rule whereby the participant would only react when the ball approached their arm. We chose the 30 cm rule (and explained to the participants) for this purpose.

R:13. Line 507-508: I'm not sure I follow the logic here. The authors propose that their embodiment manipulation worked only in the first trial, since they only observe significant

differences for their task in that time.

A: In the previous version, we only considered the first session after embodiment induction so as to observe the attentional effect during the period when we expected the effects of embodiment to be strongest. However, we have now increased the number of participants in the study, and have also included all the sessions in the analysis.

R: But that is not their measure of embodiment (questionnaires/drift). If they have no evidence from these measures to compare the different trials, it is not right to make such inferences. Even if differences in embodiment might alter behavioural outcomes, it doesn't mean that if there is no difference in behavioural outcomes, embodiment has not occurred. In addition, their questionnaires were taken at the end of all the trials. You might expect no evidence for embodiment to show itself here, if 2/3 trials were not experienced as embodied as the authors suggest. This reverse inference unfortunately comes across like cherry-picking, and quite possibly invalidates their conclusions. A follow-up experiment, with illusion inductions and embodiment measurements performed for each trial, as well as addressing the other queries I raise in my review, would help validate the findings.

A: The differences in the task behavior in the main task is not used to quantify the presence of absence of embodiment. It is used to quantify the difference in attention distribution (between the right and left hand tasks) given a higher embodiment (ie. in the EMB condition) or not (ie. the non-EMB condition).

We note again that we use all data from all the three sessions in the updated manuscript.

R:14. Lines 536-538: What is the purpose of these questions?

A: The performance perception questions (Q7-Q10) were asked to verify if the participants cognitively felt differences in performance between the two conditions.

We did not expect a change in anxiety between the conditions and Q11 and Q12 these questions may be seen as control questions in the questionnaires.

This information has been updated on lines 560-569.

Results:

R:15: Figure 2: Doesn't 'ownership' also include question 3? The authors also appear to mix up ownership, agency, and location in lines 109-110.

A: This was an error. As pointed out by the reviewer, Q 1, 2, and 3 represent ownership, and only 4 is associated with Location. We have updated Fig2 and corrected the values in Line 137-140

that refer to this result.

R:16. Lines 165-171: It is not correct to claim evidence for 'no difference' based on non-significant p-values. Bayesian analysis or equivalence testing would be appropriate.

A: We agree that the choice of wording was not appropriate here. We have also performed a Kolmogorov Smirnov test to check for the equivalence of the left hand task performance. The expression in the text has been modified to indicate this- " we could not see a difference in the left hand performance between the two conditions ($p=0.6216$ and $p=0.6216$, for the number of collisions and ball distance in Fig. 4, Kolmogorov Smirnov test)." on Line 164-165, 177-178.

R:17. Lines 184-196: Was this analysis post hoc? If so, a follow up study with the left and right hand stimuli arriving near or far in time from each other would help validate the finding.

A: Thank you for this comment. This was the post hoc test on the time bins. As a result, we found that the center time bin [-0.5 0.5] is different from all other time bins, and the time bin just before the center time bin [-1.5 -0.5] is also different from the time bins [<-1.5].

R:18. Line 205: I'm not sure why the 10 minutes is interesting. The rubber hand illusion occurs after less than 30 seconds of induction!

A: Please note that here we refer to the fast change of the reaction time and not embodiment. However, we agree that this line is maybe not very clear, or necessary here. Hence we have deleted it in the updated manuscript.

R: 19. Line 211: This correlation is not statistically significant, and should not be interpreted as such. In general, one should be very sceptical of correlations run on less than 30 participants. See this blog post for some reasons why: <https://garstats.wordpress.com/2018/06/01/smallncorr/>

A: We agree with this point and have removed this figure.

Discussion:

R: 20. Lines 230-233: Again, it is not appropriate to draw conclusions based on non-significant p-values.

A: We agree and have changed this statement in the updated manuscript.

R:21. Lines 236-237: It would be nice to have some more detailed discussion of this idea. Is attending to an embodied limb 'easier' or 'more efficient', therefore leaving more attentional resources for other tasks?

A: The key focus of the current work was to analyze and prove our hypothesis that embodiment changes the attention distribution between the limbs in a dual task. We are able to show this clearly, but our current data is unable to clarify the nature of this change. It would be very interesting to understand whether in fact less attention was utilized for the embodied arm, or whether the attention distribution was in fact ‘optimized’ (we mention these possibilities in the text). However, we are not able to clarify which was the case in our experiment. Further studies are required for clarifying this detail.

We have discussed this issue on lines 230-235 of the discussion.

R: 22. Lines 237-239: How does this account for the virtual right hand? Do the authors think this hand was also embodied?

A: Please see the answer to the reviewer’s 8th comment.

R: 23. Line 251-252: This is an incredibly strong statement and I do not believe the authors have adequate justification for it.

A: We have removed these lines in the updated discussion.

R: 24. Lines 256-257: Do their results actually provide evidence in favour of an argument for ‘hand held objects’? I’m not sure how this claim logically follows from their experiment. In both conditions an entire robot arm and hand was seen, not a hand held object. I also don’t see how their claim made in lines 260-261 follow, for similar reasons. The robot hand was not a hand held object, so they are making an inference here that is not backed up by their data.

A: We chose the example of the hand held basket (object) as we believe hand held objects come closest to what a ‘non-embodied limb’ may be in daily life scenarios. However, we agree with the reviewer that non-embodied limbs are not the same as hand-held objects and this example is further confused by the fact that some hand held objects may serve as tools, which are also considered to be ‘embodied’ (even though this embodiment differs from limb embodiment). We have thus taken the reviewer’s advice and reformulated the introduction, and now directly motivate the issue of embodiment and dual task attention, without referencing hand-held objects.

Reviewer #3 (Remarks to the Author):

Iwasaki et al studied the effect on embodiment on attention allocation in dual tasking. They employed a virtual rubber hand illusion, using asynchronous stimulation as control, and then exposed subjects to an 'easy' collision avoidance task with their left hand, and a 'difficult' cued button-press task with their right hand. There was no illusion vs non-illusion difference in task performance with respect to the left-hand collision avoidance task, but the illusion was associated with increased task score ($p=0.047$), decreased RT ($p=0.037$), and a correlation between RT change and proprioceptive drift change ($p=0.057$) for the right-hand cued button press task. The authors conclude that their results "suggest that embodiment of a limb improves attention allotment for dual task performance with it."

Although I find the idea and rationale for this study interesting, there are serious weaknesses in experimental design and the conclusions are based on just barely significant results, which make me unable to recommend this study for publication. Below are my major points.

A: We thank the reviewer for his comments. We provide the answer to his/her comments below.

- It is impossible to determine that the increase in cued-button-press task performance has anything to do with the employment of a dual task, because there is no single-task control condition. It could be the case that the observed increases in cued-button-press task performance is simply a general effect of experiencing the rubber hand illusion. In order to make inferences about the effect of embodiment on attention allocation during a dual task, the authors must conduct a control experiment with a single task (i.e., the exact same experiment but without the left-hand collision avoidance task). If the task performance effect disappears in the single-task experiment, and there is an interaction between (single-task vs dual-task) and (embodiment vs non-embodiment), one may be able to make meaningful inferences about the effect on embodiment on attention allocation during a dual task.

A: These is a misunderstanding of the purpose of the study by the reviewer. Note that in this study we were specifically interested in attention distribution (and its modulation with embodiment) *given* a dual task, and specifically we examined the effect of the embodiment on the main task performance during a dual task. We do not claim dual task employment to be the cause of the embodiment induced attention/performance effect and hence do not think it is necessary to compare the effects of embodiment on dual task performance with single task performance.

R: Why are there no control questions, to control for suggestibility and task compliance?

A: It is true that we did not have explicit control questions. However, we note that there were

several questions, and especially questions Q11 and Q12, which asked about subjects' anxieties about the task, that did not show difference between the EMB and no-EMB conditions. These suggest that there was no bias due to suggestibility or compliance between the two conditions.

R: The results are statistically weak. The effect on task score ($p=0.047$) and RT ($p=0.037$) are just barely $p<0.05$ significant, and the correlation between RT change and proprioceptive drift change is actually not ($p=0.057$). Furthermore, the correlation seems to be driven by 1 outlier showing 70 ms RT change and -40 mm drift. And the task score difference (Fig 2B) appears to be dependent on excluding one outlier (if interpret the Fig 2B correctly).

- The weak results may be due to the sample size of 16, which is very small. The authors' power calculation aimed to "provide 80% statistical power to achieve a medium effect size ($d = 0.75$)". However, $d=0.75$ corresponds to a large effect (traditionally, $d=0.5$ is considered "medium", and $d=0.8$ is considered "large"). What basis do the authors have for assuming that the effect size should be this large?

A: On the reviewer's suggestion, we have now increased the number of participants, which has enabled us to consolidate the results. In the 2-way ANOVA (Fig. 5) which is the key result supporting our conclusion, we see main effects with significance of $p<10^{-13}$ ($\eta^2=0.281$) and $p<0.005$ ($\eta^2=0.037$). We also observe stronger differences in our T-test/Wilcoxon test value on embodiment scores (ownership: $Z(23)=-3.947$, $p<0.0001$; location: $Z(23)=-3.584$, $p<0.0001$, and agency: $Z(23)=-4.259$, $p<0.0001$ respectively) and proprioceptive drift ($Z(23)=-3.857$, $p<0.001$, $d=1.1897$) between EMB and no-EMB conditions.

R: Fig 2 states that Q3 ("The robot arm belongs to you") is a "location" statement. However, this statement is clearly an ownership statement and should be analyzed as that.

- The results section: "We observed that the ownership (average score in Q1 and Q2), agency (average of Q3 and Q4) as well the sense of location (average of Q5 and Q6) towards the robot arm was [...]". There is some serious confusion about the numbering of the statements here. In the methods section, agency statements are Q5 & Q6, not Q3 & Q4. Also, as stated in the previous point, Q3 ("The robot arm belongs to you") cannot be regarded and analyzed as a "location" statement - this is an ownership statement.

A: This was an error. As pointed out by the reviewer, Q 1, 2, and 3 represent ownership, and only 4 is associated with Location. We have updated Fig2 and corrected the values in Line 137-140 that refer to this result.

R: Throughout the manuscript, the authors use the grocery shop basket as an example of everyday life dual tasking. They state: "Ownership of a limb is known to improve its visual awareness

(Hoort, Reingardt, and Ehrsson, 2017), suggesting that we will be more aware of obstacles to our swinging arm, than to the swinging basket in our arms, which we do not feel a sense of embodiment towards.” This notion is not necessarily true. A basket held in one’s hand should be considered as tool, which, in many respects, have been shown to be incorporated into the body image (even though people don’t have ownership feelings over tools). The study by Hoort et al compared the visual awareness of an owned rubber hand versus an un-owned rubber hand, which has no implications for the awareness of one’s own body versus a tool (because an ‘un-owned’ rubber hand is not a tool.)

A: We chose the example of the hand held basket (object) as we believe they come closest to what a ‘non-embodied limb’ may be in daily life scenarios. However, we agree with the reviewer that non-embodied limbs are not the same as hand-held objects and this example is further confused by the fact that some hand held objects may serve as tools, which are also considered to be ‘embodied’. We have thus reformulated the introduction, and now directly motivate the issue of embodiment and dual task attention, without referencing hand-held objects.

R:- The language throughout the paper needs to be improved. I would recommend a professional language service.

A: We have had the updated manuscript read by a native English speaker.

Minor:

R: Abstract: line 21. “don’t” -> ”did not”

A: Thank you. We have corrected the typo.

R: Line 141. “let” -> left.

A: We have corrected the typo.

R: Please define what a “ca-button” is.

A: ca-button is defined in the results on lines 106-110

“As their secondary task, the participants were required to press a collision avoidance button (ca-button) with their left thumb on the handle of the haptic device held in their left hand when they perceived a danger of collision. The ca-button press resulted in the ball being deflected away from their hand. The participant were asked not to move their left arm during the task.”

We have now further described it in the methods on line 514-522.

R: Line 160. “cm” -> “m”

A: Thank you. corrected.

R: Line 190. “ $p < 10^{-8}$ ” - A number is lacking.

Throughout the manuscript, exact p values are often described as “ $p < 0.XX$ ”, when it should say “ $p = 0.XX$ ”.

A: This was done to account for the rounding off of the p value for some statistics. We have re-verified all the p-values to ensure there is not error in the reporting.

REVIEWERS' COMMENTS:

Reviewer #1 (Remarks to the Author):

All my concerns have been well addressed. The manuscript has been appreciably improved.

Reviewer #2 (Remarks to the Author):

The authors have adequately addressed my comments. I only have a few minor points that could be fixed at the proofing stage:

There is a typo in 'proprioceptive drift' in Figure 2.

It needs to be stated which direction the robot hand was shifted in relative to the real hand. The authors state 'towards the body' - I assume this is towards the body midline (i.e., from left to right, rather than a movement backwards).

Line 178: the two p-values listed here are the same. The authors may need to double check one of them is not incorrect.

Please provide a data accessibility statement.

We thank the reviewers and editor for their time and constructive comments that have contributed to greatly improve the manuscript. Below we provide our final updates for each of the reviewer comment. The corresponding major changes in the manuscript have been highlighted in the “changes_underlined_manuscript.docx” file.

Reviewer #1 (Remarks to the Author):

R: All my concerns have been well addressed. The manuscript has been appreciably improved.

We thank the reviewer for his/her time and effort to improve our manuscripts.

Reviewer #2 (Remarks to the Author):

R: The authors have adequately addressed my comments. I only have a few minor points that could be fixed at the proofing stage:

We thank the reviewer for his/her comments to improve our manuscripts.

R: There is a typo in 'proprioceptive drift' in Figure 2.

A: We have found this typo in Figure 3. Thank the reviewer for pointing out this.

R: It needs to be stated which direction the robot hand was shifted in relative to the real hand. The authors state 'towards the body' - I assume this is towards the body midline (i.e., from left to right, rather than a movement backwards).

A: We have added the explanation to Line 80 (The robot arm was purposely presented displaced, by 10 cm horizontally towards the body midline, from the real arm) and Line 331 (When the VR image reappeared, the robot arm was horizontally shifted 10 cm, without their knowledge, towards the participant's body midline).

R: Line 178: the two p-values listed here are the same. The authors may need to double check one of them is not incorrect.

A: We recalculated these values, and confirm the value is correct.

R: Please provide a data accessibility statement.

A: We have added the “Data availability” and “Code availability” section on Line 470-473 and 476-478.